# Decline of PD-L1 Immunoreactivity with Storage Duration in Formalin-Fixed Paraffin-Embedded Breast Cancer Specimens: Implications for Diagnostic Accuracy and Immunotherapy Eligibility in Triple-Negative Breast Cancer

**DOI:** 10.3390/cancers17193103

**Published:** 2025-09-23

**Authors:** Keiko Yanagihara, Koji Nagata, Tamami Yamakawa, Sena Kato, Miki Tamura, Masato Yoshida

**Affiliations:** 1Department of Breast Surgery and Oncology, Nippon Medical School Tama-Nagayama Hospital, 1-7-1 Nagayama, Tama-shi 206-8512, Tokyo, Japan; s13-102yt@nms.ac.jp (T.Y.); s14-031ks@nms.ac.jp (S.K.); t-miki@nms.ac.jp (M.T.); 2Department of Pathology, Nippon Medical School Tama-Nagayama Hospital, 1-7-1 Nagayama, Tama-shi 206-8512, Tokyo, Japan; k-nagata@nms.ac.jp; 3Department of Pharmacy, Nippon Medical School Tama-Nagayama Hospital, 1-7-1 Nagayama, Tama-shi 206-8512, Tokyo, Japan; yoshida-m@nms.ac.jp

**Keywords:** PD-L1, breast cancer, immune checkpoint inhibitor, triple negative breast cancer, immunohistochemistry

## Abstract

This study examined how long-term storage of formalin-fixed paraffin-embedded (FFPE) breast cancer tissue affects PD-L1 testing, a key biomarker for immunotherapy eligibility in triple-negative breast cancer. We analyzed 63 cases and found that PD-L1 staining significantly declined when blocks were stored for three or more years, potentially leading to false-negative results. PD-L1 positivity was linked to higher tumor grade and Ki67 proliferation. Our findings highlight the importance of using recent tissue for PD-L1 testing to ensure accurate diagnosis and optimal immune checkpoint inhibitor treatment selection.

## 1. Introduction

Breast cancer remains the most common malignancy and the leading cause of cancer-related mortality among women worldwide [1]. In 2020 alone, more than 2.3 million women were diagnosed with breast cancer, and approximately 685,000 women died from the disease globally. While advances in early detection and targeted therapy have significantly improved outcomes for many breast cancer subtypes [2,3], prognosis and recurrence risk vary markedly according to molecular subtype [4,5]. This classification is based on the presence or absence of hormone receptors—estrogen receptor (ER) and progesterone receptor (PgR)—and the amplification or overexpression of human epidermal growth factor receptor 2 (HER2). Triple-negative breast cancer (TNBC) is defined by the absence of ER and PgR expression and the lack of HER2 overexpression or amplification. TNBC accounts for approximately 15–20% of all breast cancers, both globally and in Japan [6,7], and is disproportionately associated with younger age at diagnosis, higher histologic grade, and more aggressive clinical behavior [8]. TNBC is characterized by early recurrence, frequent visceral and central nervous system metastases, and overall poorer survival compared with other subtypes [9,10,11].

A systematic review by Morgan et al. (2024), involving over 280,000 breast cancer patients, demonstrated that hormone receptor–negative disease—including TNBC—carries a significantly higher risk of recurrence than hormone receptor–positive subtypes, regardless of follow-up duration [9]. The biological aggressiveness of TNBC is reflected in its higher proliferative index, frequent *TP53* mutations, and enrichment for basal-like gene expression profiles [8,11]. Unlike hormone receptor-positive breast cancers, TNBC lacks well-established targeted therapies, leaving chemotherapy as the mainstay of treatment for many years [9,10]. In the CALGB9741 trial, dose-dense chemotherapy administered every two weeks demonstrated significantly greater efficacy compared to chemotherapy administered every three weeks in premenopausal women, showing a 7% absolute improvement in 3-year disease-free survival (DFS) and a 2% improvement in overall survival (OS), indicating a significant increase in survival rates. However, recurrence rates remain high [12].

Recent advances in cancer immunology have introduced immune checkpoint inhibitors (ICIs) as a promising therapeutic option in TNBC [13,14,15]. These agents target inhibitory pathways, such as the programmed cell death-1 (PD-1)/programmed death-ligand 1 (PD-L1) axis, and restore anti-tumor immunity. PD-L1 is expressed on tumor cells and tumor-infiltrating immune cells, serving as a critical immune checkpoint molecule. Its binding to PD-1 receptors on T cells suppresses cytokine release and proliferation, enabling tumor immune escape [16]. In TNBC, PD-L1 expression assessed by immunohistochemistry (IHC) is a predictive biomarker for ICI efficacy [17]. The phase III KEYNOTE-355 trial found pembrolizumab plus chemotherapy significantly prolonged progression-free survival (PFS) in PD-L1 positive metastatic TNBC, defined by a Combined Positive Score (CPS) ≥ 10 using the 22C3 pharmDx assay [18]. Similarly, the IMpassion130 trial found enhanced PFS with atezolizumab plus nab-paclitaxel in PD-L1 positive TNBC, determined by the SP142 assay [15]. These data have made PD-L1 testing a routine requirement for metastatic TNBC, directly influencing ICI eligibility [13,14,15].

However, PD-L1 testing accuracy is vulnerable to variability across pre-analytical, analytical, and post-analytical stages. Pre-analytical factors—such as cold ischemia time, fixation method/duration, fixative type, and processing protocols—significantly influence antigen preservation [18]. Among these, storage conditions of formalin-fixed paraffin-embedded (FFPE) blocks remain underappreciated but critically important for assay fidelity. FFPE blocks are commonly archived and used for retrospective studies, often stored for years. Yet, evidence suggests prolonged storage may lead to antigen degradation, resulting in reduced immunoreactivity and potential false-negative IHC results [19,20,21,22].

Grillo et al. (2017) documented significant declines in immunoreactivity for multiple markers after >5 years storage [19], while Engel and Moore (2011) underscored the sensitivity of antigenic epitopes to pre-analytical variables such as fixation and storage [18]. A false-negative PD-L1 result due to storage degradation is particularly concerning in the context of ICI eligibility (e.g., CPS ≥ 10 in KEYNOTE-355) [14]. The causes of false negatives are likely protein oxidation, hydrolysis, cross-linking, paraffin embedding effects, and environmental factors like humidity and temperature [23,24]. Controlled low-temperature, low-humidity storage of slides has been suggested to mitigate such loss.

This issue is especially relevant for PD-L1 as a predictive marker [25]. False negatives may inappropriately exclude patients from ICI treatment. In such cases, patients may face life-threatening situations at an early stage. If false negatives increase, many patients will be affected by this outcome. Although previous studies reported storage-related degradation of markers like ER, PgR, Ki67, and HER2 [19,20,21,22], data specific to PD-L1 stability in archived FFPE, especially in TNBC, remain sparse and inconsistent.

Furthermore, PD-L1 positivity is associated with infiltrating lymphocytes and heightened proliferation [26,27]. Schalper et al. (2014) associated PD-L1 mRNA with inflamed microenvironments and better outcomes [27]. PD-L1 thus acts as both a prognostic and predictive marker [14,27,28].

Most pathology labs store FFPE blocks long-term. Current ASCO/CAP guidelines do not specify storage limits [29]

We hypothesized that prolonged storage of FFPE TNBC samples reduces PD-L1 immunoreactivity (22C3 pharmDx assay). To test this, we retrospectively analyzed TNBC cases with PD-L1 testing at diagnosis and repeated testing of the same blocks after storage intervals (<1 year, 1–2 years, 2–3 years, ≥3 years). We examined associations with clinicopathologic features, pathologic complete response (pCR) post-neoadjuvant chemotherapy (NAC), and survival outcomes.

## 2. Materials and Methods

### 2.1. Study Population

We retrospectively reviewed 63 cases diagnosed as TNBC at our institution between 1 April 2020 and 31 March 2024, in which PD-L1 expression was evaluated using the 22C3 antibody clone at the time of diagnosis. Only patients with available FFPE tissue blocks stored at room temperature were included.

### 2.2. Pathological Assessment at Diagnosis

At the time of diagnosis, pathological parameters including PD-L1 (22C3) status, nuclear grade (NG), ER, PgR, HER2, Ki67 proliferation index, and *p53* mutation status were assessed based on routine pathology reports. Re-examination of original slides was not performed for this study, as prolonged storage of unstained slides can also result in decreased immunoreactivity [7,8].

### 2.3. FFPE Storage and Restaining

For evaluation of post-storage PD-L1 status, the original FFPE blocks stored at room temperature were thinly sliced, and new slides were prepared for PD-L1 IHC using the same 22C3 clone.

### 2.4. Tissue Fixation and Processing

Specimens from core needle biopsies or surgical resections were fixed in 10% neutral buffered formalin for 10–72 h prior to paraffin embedding.

### 2.5. PD-L1 Immunohistochemistry (22C3 pharmDx)

PD-L1 staining was performed using the PD-L1 IHC 22C3 pharmDx kit (Agilent Technologies, Santa Clara, CA, USA) in accordance with the manufacturer’s instructions. Four-micrometer-thick FFPE tissue sections were deparaffinized and processed on the Dako Autostainer Link 48 platform. Antigen retrieval was performed using EnVision FLEX Target Retrieval Solution (Low pH). Endogenous peroxidase activity was quenched, and sections were incubated with the mouse monoclonal anti–PD-L1 antibody (clone 22C3). Visualization was achieved using the EnVision FLEX visualization system, followed by hematoxylin counterstaining.

PD-L1 expression was quantified using the Combined Positive Score (CPS). The CPS is calculated as the number of PD-L1–positive cells, including tumor cells, lymphocytes, and macrophages, divided by the total number of viable tumor cells, multiplied by 100.CPS = (Number of PD-L1–positive cells [tumor cells, lymphocytes, macrophages] × 100)Total number of viable tumor cells

CPS ≥ 10 was considered positive, according to current guidelines [6].

### 2.6. Classification of Storage Duration

FFPE storage duration was categorized as:
<1 year1–2 years2–3 years≥3 years
PD-L1 status changes were classified as:
Increased stainingNo changeDecreased staining (CPS decreased to <10)


### 2.7. Association with Clinicopathologic Factors

We examined correlations between PD-L1 changes and Ki67, NG, *p53* mutation status, and pathological complete response (pCR) rate among NAC patients. pCR was defined as the complete disappearance of invasive carcinoma in the breast, irrespective of residual ductal carcinoma in situ.

### 2.8. Survival Analysis

For patients with stage I–III disease, recurrence-free survival (RFS) and overall survival (OS) were calculated from the date of diagnosis to the date of recurrence, death, or last follow-up. Survival curves were generated using the Kaplan–Meier method.

### 2.9. Statistical Analysis

Differences between categorical variables were analyzed using the chi-square test or Fisher’s exact test, as appropriate. Logistic regression was used to explore associations between PD-L1 decline and clinicopathologic factors. Kaplan–Meier curves were compared using the log-rank test. A *p*-value < 0.05 was considered statistically significant.

## 3. Results

### 3.1. Patient Characteristics

A total of 63 TNBC patients met the inclusion criteria. At diagnosis, 22 patients (34.9%) were PD-L1–negative (CPS < 10) and 41 patients (65.1%) were PD-L1–positive (CPS ≥ 10). Table 1 summarizes the clinicopathologic characteristics.

Among PD-L1–negative cases, storage duration was ≥3 years in 9 patients (41%), 2–3 years in 9 (41%), 1–2 years in 2 (9%), and <1 year in 2 (9%). In PD-L1–positive cases, the respective distribution was 8 (20%), 16 (39%), 9 (22%), and 8 (20%).

Median Ki67 was significantly higher in PD-L1–positive cases than in PD-L1–negative cases (48.2% [20–99%] vs. 25.2% [7–90%], *p* = 0.00054). Nuclear grade was also significantly higher in PD-L1–positive cases, with grade 3 in 26 patients (63%) vs. grade 1 in 9 patients (41%) among PD-L1–negative cases (*p* = 0.0381). No significant association was observed between PD-L1 status and *p53* mutation type (*p* = 0.075).

At diagnosis, distant metastases were present in three PD-L1–negative patients (14%) and four PD-L1–positive patients (10%). During follow-up, distant recurrence occurred in five PD-L1–positive patients (14%), of whom three (8%) died. No recurrences or deaths occurred in the PD-L1–negative group.

### 3.2. Neoadjuvant Chemotherapy and pCR

NAC was administered to 11 PD-L1–negative patients (50%) and 15 PD-L1–positive patients (37%). Among PD-L1–positive patients, pCR was achieved in five of 15 (33%), compared to none in the PD-L1–negative group. The difference approached but did not reach statistical significance (*p* = 0.0527) (Table 2). The results of chemotherapy regimens and treatment effects according to PD-L1 expression are shown in Table 3. There were few ICI regimens and many dose-dense regimens. Although the number of patients in the ICI regimen was small (two patients), pCR was achieved in both cases. Additionally, both patients were PD-L1 positive.

### 3.3. Change in PD-L1 Status After Storage

No cases showed increased PD-L1 staining after storage. In PD-L1–positive patients, decreased staining occurred in 0% of cases stored < 1 year, 11% (1/9) stored 1–2 years, 13% (2/16) stored 2–3 years, and 50% (4/8) stored ≥ 3 years. Figure 1 shows an example of the change in three years after storage. Compared to the original, staining intensity was clearly reduced in the specimen after three years of storage. The association between longer storage duration and PD-L1 decline was statistically significant (*p* = 0.015). In PD-L1–negative patients, no change in staining was observed regardless of storage duration (Table 4 and Table 5).

### 3.4. Survival Outcomes

Recurrence was observed in five PD-L1-positive cases. Of these, three patients had died. No recurrence or deaths were observed in PD-L1-negative cases. Due to the limited number of events, survival analysis did not reach statistical significance; however, a trend toward shorter RFS (*p* = 0.096, log-rank test) and OS (*p* = 0.183, log-rank test) was observed in PD-L1–positive patients. The RFS curve is shown in Figure 2. Due to the small number of events, no difference based on PD-L1 expression was demonstrated.

## 4. Discussion

Triple-negative breast cancer (TNBC) is defined by the absence of ER and PgR expression and the lack of HER2 overexpression or amplification. TNBC accounts for approximately 15–20% of all breast cancers, both globally and in Japan [6,7], and is disproportionately associated with younger age at diagnosis, higher histologic grade, and more aggressive clinical behavior [8]. TNBC is characterized by early recurrence, frequent visceral and central nervous system metastases, and overall poorer survival compared with other subtypes [1,2,3]. Recent advances in cancer immunology have introduced immune checkpoint inhibitors (ICIs) as a promising therapeutic option in TNBC [12,13,14,15].

In TNBC, PD-L1 expression assessed by immunohistochemistry (IHC) is a predictive biomarker for ICI efficacy [16]. Therefore, confirming the presence or absence of PD-L1 expression is necessary for the appropriate selection of ICI therapy.

When assessing PD-L1 at recurrence, FFPE blocks stored for several years are often used. However, these FFPE blocks may yield false negatives due to reduced PD-L1 expression. Consequently, this could lead to the omission of ICI regimens, potentially compromising patient survival.

We investigated the association between the storage duration of FFPE blocks and PD-L1 false negatives. We believed that if we could demonstrate that long-term storage may lead to false negatives, it would prompt caution regarding the use of aged FFPE blocks as companion diagnostics for ICI therapy.

In this comprehensive retrospective analysis, we demonstrated that prolonged storage of FFPE TNBC specimens—particularly beyond three years—is significantly associated with reduced PD-L1 immunoreactivity using the 22C3 pharmDx assay. Approximately 50% of long-stored samples exhibited decreased staining compared to no decline in samples stored for less than one year, and no cases showed increased positivity post-storage. Regarding associations with other pathological markers, PD-L1 positivity at diagnosis correlated with higher Ki-67 proliferation indices and nuclear grade but not with *p53* status. While PD-L1 positivity appeared to be associated with higher pathologic complete response rates after neoadjuvant chemotherapy, the difference did not reach statistical significance, likely due to limited sample size. Particularly in PD-L1-positive cases, treatment efficacy with ICI regimens is especially anticipated. However, only two cases actually received ICI therapy, and this did not lead to a significant increase in the pCR rate. Survival outcomes trended worse in PD-L1–positive patients, although statistical significance was not achieved, again due to the small number of events.

Chemotherapy-only datasets consistently show that baseline PD-L1 positivity associates with higher pathologic complete response (pCR) in early triple-negative breast cancer (TNBC). In the largest TNBC-specific meta-analysis (19 studies; *n* = 2403), PD-L1–positive tumors had nearly double the odds of pCR to standard neoadjuvant chemotherapy versus PD-L1–negative tumors (pooled OR 1.95, 95% CI 1.39–2.73), with favorable effects also seen for DFS and OS—supporting PD-L1 as a correlate of chemosensitivity rather than a direct therapeutic target in this context [30]. Concordant single-center data using quantitative IHC likewise found that pretreatment epithelial or stromal PD-L1 independently predicted pCR to anthracycline/taxane-based regimens, though the signal overlaps with tumor-infiltrating lymphocytes (TILs) [31]. Broader breast-cancer meta-analysis reinforces the trend (OR ≈ 2.0 for pCR with PD-L1 positivity), albeit with heterogeneous assays and cut-offs. Notably, PD-L1 is dynamic and can shift after neoadjuvant therapy, underscoring the importance of evaluating pretreatment biopsies with validated assays (e.g., 22C3/CPS or SP142/IC) when interpreting predictive associations.

Our findings corroborate previous reports indicating antigen degradation in long-stored FFPE tissue blocks. Grillo et al. (2017) showed substantial loss of immunoreactivity for markers including hormone receptors and Ki-67 in blocks stored for more than five years [19,20]. In particular, they state that while cytoplasmic antigens showed little degradation, membrane and nuclear antigens exhibited reduced immunostaining intensity in older blocks. Deep sectioning combined with extended antigen retrieval (AR) has been reported to partially restore antigenicity, but its accuracy has limitations. Therefore, when performing immunohistochemical testing using old FFPE blocks, the risk of false negatives or reduced expression should be noted as a limitation. Engel and Moore (2011) highlighted how pre-analytical variables—such as fixation duration and storage conditions—critically influence protein epitope preservation [18]. Our results extend these observations specifically to PD-L1, a marker of escalating therapeutic importance.

Mechanistically, antigen degradation in FFPE specimens is thought to result from oxidative and hydrolytic modifications, cross-linking of amino acid residues, and conformational changes that mask epitopes, all exacerbated by environmental factors like humidity and temperature fluctuation [23,24]. It is recommended to maintain FFPE blocks below 27 °C and at a humidity level below 30–70%.

From a clinical standpoint, false-negative PD-L1 results due to storage-related degradation could misclassify eligible patients as ineligible for immune checkpoint inhibitors, such as pembrolizumab in metastatic TNBC (KEYNOTE-355 criteria: CPS ≥ 10) [14]. This underscores the urgency to prioritize testing on recent tissue whenever possible.

Beyond tissue blocks, emerging data also document antigen decay in mounted unstained sections. He et al. (2023) studied the effect of storage time and temperature on PD-L1 (SP142) in invasive breast cancer sections and found that positivity dropped from 97.2% at 1 week to 33% at 24 weeks at room temperature; refrigerated storage (4 °C or −20 °C) delayed but did not prevent this decline [32]. Additionally, Fernández et al. (2023) compared 22C3 (ECD-binding) against E1L3N (ICD-binding) antibodies and found archival samples lost signal significantly in 22C3 but less so in E1L3N, implying epitope specificity affects degradation [33]. These reinforce the perils of delaying both staining and slide processing post-sectioning.

Accelerated instability testing by Haragan et al. (2020) revealed that the PD-L1 IHC signal diminishes markedly over time for 22C3, 28–8, and SP142, whereas mass spectrometry detected the PD-L1 protein as intact, suggesting structural epitope distortion—not protein degradation—is the driver, and that desiccant storage may mitigate loss [34].

The clinical relevance of PD-L1 extends beyond TNBC. PD-L1 expression correlates with aggressive tumor biology, including high nuclear grade, higher Ki-67 rates and basal-like features, as we observed [26,27]. PD-L1 expression is also associated with tumor-infiltrating lymphocytes and an inflamed microenvironment—features linked to better prognosis in certain contexts, highlighting the dual prognostic and predictive roles of PD-L1.

Current ASCO/CAP guidelines do not specify an upper storage limit for FFPE blocks in PD-L1 testing [29], but our data and supporting studies suggest practical recommendations.

Use the most recent tissue block possible. If using older FFPE blocks, record the storage duration. If only old tissue is available, consider repeating the biopsy. Optimize storage conditions, minimize humidity and temperature fluctuations. Process unstained slides promptly to accelerate staining and reduce antigen loss.

Furthermore, clinicians must recognize that immunohistopathological staining using older FFPE blocks may yield false negatives. This implies limitations in accuracy as a companion diagnostic, and appropriate therapeutic selection must not be compromised.

Limitations include the single-institution design, small size of long-storage cohorts, absence of precise storage condition data, and exclusive use of 22C3 assay, as different PD-L1 clones may behave differently.

Future multicenter studies should validate PD-L1 stability across various assays and storage protocols, and explore advanced antigen retrieval, mass spectrometry, or desalting techniques to rescue signals from degraded specimens. Quality assurance should incorporate storage duration as a pivotal pre-analytical variable in predictive biomarker evaluation.

## 5. Conclusions

PD-L1 immunoreactivity in TNBC declines significantly after ≥3 years of FFPE storage, risking false-negative results and missed immunotherapy opportunities. If ICI therapy is not appropriately selected, it can be life-threatening for the patient. Prioritizing recent specimens and stricter pre-analytical protocols will help maintain biomarker integrity in clinical practice.

## Figures and Tables

**Figure 1 cancers-17-03103-f001:**
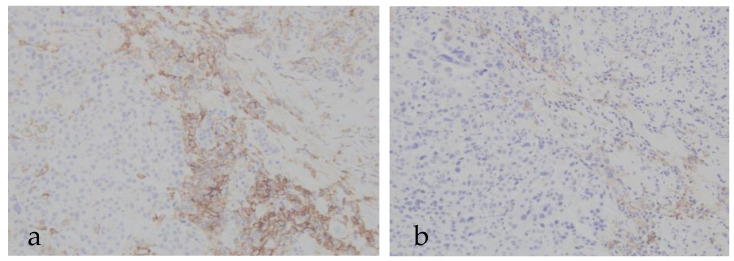
Changes in PD-L1 staining. (**a**) Original staining without storage period. (**b**) Re-stained FFPE specimen stored for three years.

**Figure 2 cancers-17-03103-f002:**
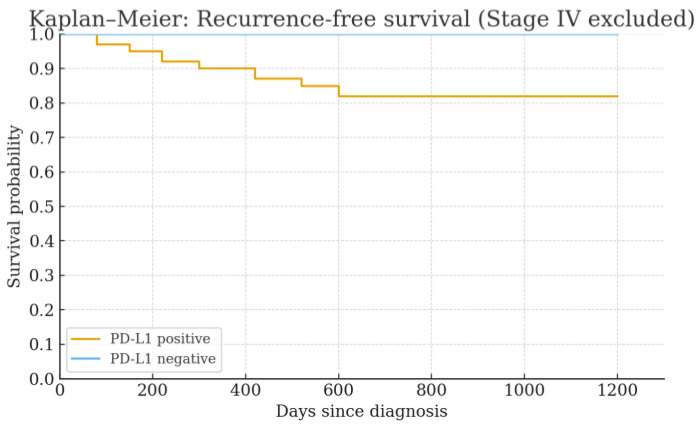
Recurrence-free survival curves for PD-L1-positive and PD-L1-negative patients.

**Table 1 cancers-17-03103-t001:** Comparison of clinical and pathological features by PD-L1 expression status.

	Negative	Positive
Number of cases (*n*)	22	41
Age (years) (mean)	42–87 (63.3)	42–93 (67.4)
Observation period (days) (median)	94–1334 (869)	63–1298 (574)
FFPE elapsed years ≥ 3 (cases)	9 (41%)	8 (20%)
2–3 years	9 (41%)	16 (39%)
1–2 years	2 (9%)	9 (22%)
<1 year	2 (9%)	8 (20%)
Ki67 (median)	7–90 (25.2%)	20–99 (48.2%)
NG 1	9 (41%)	6 (15%)
2	6 (27%)	8 (20%)
3	6 (27%)	26 (63%)
Unknown	1 (5%)	1 (2%)
*p53* wild type	11 (50%)	15 (37%)
mutant	7 (32%)	20 (49%)
null	0	3 (7%)
equivocal	2 (9%)	0
unknown	2 (9%)	3 (7%)
Stage I	11 (50%)	8 (20%)
II	6 (27%)	20 (49%)
III	2 (9%)	9 (22%)
IV	3 (14%)	4 (10%)
Neoadjuvant chemotherapy: Yes	11 (50%)	15 (37%)
No	11 (50%)	26 ** (63%)
Number of recurrence cases *	0	5 (14%)
Number of deaths *	0	3 (8%)

* Excluding cases diagnosed as stage IV at initial visit. ** Including two cases undergoing preoperative chemotherapy.

**Table 2 cancers-17-03103-t002:** PD-L1 status and pathological treatment efficacy.

	Negative (*n* = 11)	Positive (*n* = 15)	Fisher’s Exact Test
pCR	0 (0%)	5 (33%)	*p*-value
non-pCR	11 (100%)	10 (67%)	0.0527

**Table 3 cancers-17-03103-t003:** Chemotherapy regimens and treatment effects according to PD-L1 status.

	Negative (*n* = 11)			Positive (*n* = 15)		
Regimen	*n* = 11	non-pCR	pCR	*n* = 15	non-pCR	pCR
Dose Dense	8	8	0	9	6	3
Triweekly	2	2	0	1	1	0
ICI combination	0	0	0	2	0	2
Bevacizumab + Paclitaxel	1	1	0	3	3	0

**Table 4 cancers-17-03103-t004:** PD-L1 status after storage according to PD-L1 status original PD-L1 status.

	Negative (*n* = 22)	Positive (*n* = 41)
Increased staining	0 (0%)	0 (0%)
No change	22 (100%)	34 (83%)
Decreased staining	0 (0%)	7 (17%)

**Table 5 cancers-17-03103-t005:** Change in PD-L1 status after storage.

	Negative (*n* = 22)			Positive (*n* = 41)			
After Storage	*n*	No Change	Decreased	*n*	No Change	Decreased	Negative vs. Positive
<3 years	9	9 (100%)	0	8	4 (50%)	4 (50%)	*p* = 0.0294
2–3 years	9	9 (100%)	0	16	14 (88%)	2 (13%)	*p* = 0.520
1–2 years	2	2 (100%)	0	9	8 (89%)	1 (11%)	*p* = 1.000
>1 year	2	2 (100%)	0	8	8 (100%)	0	*p* = 1.000
Total	22	22 (100%)	0	41	34 (83%)	7 (17%)	
	N/A			*p* = 0.015			Cochran-Armitage trend test

## Data Availability

The data presented in this study are available on request from the corresponding author.

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
