# Peer review of "Decline of PD-L1 Immunoreactivity with Storage Duration in Formalin-Fixed Paraffin-Embedded Breast Cancer Specimens: Implications for Diagnostic Accuracy and Immunotherapy Eligibility in Triple-Negative Breast Cancer"

_cancers, 2025, doi:10.3390/cancers17193103_

Round 1
Reviewer 1 Report
Comments and Suggestions for Authors
Manuscript Title: Decline of PD-L1 Immunoreactivity with Storage Duration in Formalin-Fixed Paraffin-Embedded Breast Cancer Specimens: Implications for Diagnostic Accuracy and Immunotherapy Eligibility in Triple-Negative Breast Cancer
Manuscript ID: cancers-3857919
The authors have very well explained the overall information about the research article is good. Now a days Breast cancer among woman is really a threat to the community and is so relevant for patients receive immunotherapy. Still some minor correction is needed:
However, a few concerns persist.
- The author describes in summary part line no. 22-27 that the long-term formalin-fixed paraffin embedded breast cancer tissue affect PDL 1 staining. So, the author asked to clarify what is the necessity for formalin-fixed paraffin embedded breast cancer tissue storage for cancer diagnosis or treatment.
- Please mention the sample size in the section of Materials and Methods, although it is mentioned in abstract part and results part.
- Provide a figure of CONSORT diagram which indicates disposition of the study participants.
- In Results section it was observed that Line 192-199, the author describes qPCR data as percentage, but it may be more predictable is the data presented through Agarose Gel Electrophoresis. Please specify.
- In line no 137-147, Image of tissue section of PDL 1 Immunohistochemistry staining after storage should be given as a separate figure to show the immunoreactivity was declined or not.
- Page 2, line 63: please add the year of cross reference, i.e. Morgan et al. (2024). Similarly add year in cross reference in the line number 93, 106, 229.
- Page 6, line 213: remove extra full stop.
- Table 5. In the section of Negative, correct the spelling “After strage”.
- Page 8, line 269 to 271 make a single sentence.
- Please modify the reference no 1, 8, 23 in the main reference section. Abbreviated Journal Name should be recommended. Follow the style of references according to author guidelines and remove the extra number in reference numbering (i.e. 2,3,4,5,6,7,8….).
Author Response
To Reviewer 1
Thank you very much for your valuable comments. We have revised the manuscript in accordance with your suggestions.
<Point 1> The author describes in summary part line no. 22-27 that the long-term formalin-fixed paraffin embedded breast cancer tissue affect PDL 1 staining. So, the author asked to clarify what is the necessity for formalin-fixed paraffin embedded breast cancer tissue storage for cancer diagnosis or treatment.
<Response> As noted in lines 246 to 250, it is necessary to identify predictive factors for efficacy and select treatment drugs even in recurrent breast cancer. While a rebiopsy of the recurrent site would be ideal, this is typically difficult. Therefore, it is necessary to examine treatment predictive factors using previously created and stored FFPE blocks, suggesting that long-term preservation of FFPE is essential.
<Point 2> Please mention the sample size in the section of Materials and Methods, although it is mentioned in abstract part and results part.
<Response> I have filled it out.
<Point 3> Provide a figure of CONSORT diagram which indicates disposition of the study participants.
<Response> This study is a simple classification only and lacks progression, so I did not include it as it might cause confusion.
<Point 4> In Results section it was observed that Line 192-199, the author describes qPCR data as percentage, but it may be more predictable is the data presented through Agarose Gel Electrophoresis. Please specify.
<Response> we would like to clarify that in this study pCR denotes pathological complete response. Specifically, it refers to the proportion of patients who received neoadjuvant chemotherapy and demonstrated complete disappearance of cancer cells in the postoperative pathological examination, analyzed according to PD-L1 status. As this definition is distinct from the qPCR data you referred to, we did not introduce revisions on that particular point.
<Point 5> In line no 137-147, Image of tissue section of PDL 1 Immunohistochemistry staining after storage should be given as a separate figure to show the immunoreactivity was declined or not.
<Response> Figure 1 shows the difference in PD-L1 antibody staining between specimens stained immediately after biopsy and those prepared from FFPE blocks stored for over three years.
<Point 6> Page 2, line 63: please add the year of cross reference, i.e. Morgan et al. (2024). Similarly add year in cross reference in the line number 93, 106, 229.
<Response> Thank you for pointing that out. I have added the year.
<Point 7> Page 6, line 213: remove extra full stop.
<Response> We have revised the description of survival outcomes.
<Point 8> Table 5. In the section of Negative, correct the spelling “After strage”.
<Response> I have made the correction.
<Point 9> Page 8, line 269 to 271 make a single sentence.
<Response> I have made the correction.
<Point 10> Please modify the reference no 1, 8, 23 in the main reference section. Abbreviated Journal Name should be recommended. Follow the style of references according to author guidelines and remove the extra number in reference numbering (i.e. 2,3,4,5,6,7,8….).
<Response> Thank you for pointing that out. It seems the file was converted during saving. I've fixed it.

Reviewer 2 Report
Comments and Suggestions for Authors
This study investigates the impact of FFPE storage time on PD-L1 immunoreactivity. The following issues need to be addressed before publication.
Line 36: Please remove "No increased staining occurred." The meaning of this sentence is redundant with the previous one.
Line 69: Please emphasize the limitations of chemotherapy and use this to highlight the value of ICIs.
Introduction: You mentioned the studies by Grillo and Engel, which demonstrated a decline in the immunoreactivity of markers with prolonged storage time. Please emphasize the adverse consequences of this decline in immunoreactivity for these markers, such as how many patients might be misdiagnosed as a result.
Line 119: Please add the number of cases.
Line 202: It is quite rare to see a table cited in a title. Please correct this.
Please change the "p" used to indicate significance to italics.
Please add spaces before and after the equals sign.
Please check the formatting of the table text, such as capitalizing the first letter of "total."
Figure 1: The figure caption is not specific enough. Please provide a detailed description of what this survival curve represents.
Please change the font of the figure to Times New Roman.
Please review the entire manuscript, as I noticed some punctuation marks are not used correctly, particularly the frequent omission of periods at the end of paragraphs.
Lines 223-227: The meanings of these two sentences are similar. Please delete one of them.
Overall, the clinical significance of this paper is relatively limited. Could you elaborate on the clinical implications of this study from a more in-depth perspective?
Author Response
To Reviewer 2
Thank you very much for your thoughtful comments. We have made the revisions.
We respectfully emphasize that the key clinical implication of our study is the potential risk of false-negative PD-L1 results when testing is performed at the time of recurrence using surgical specimens obtained several years earlier. This issue is of particular concern in triple-negative breast cancer, where treatment options remain limited. In such cases, patients who might otherwise be eligible for immune checkpoint inhibitors (ICIs) could be denied access to this therapy due to false-negative PD-L1 findings, which may ultimately lead to worse survival outcomes.
<Point 1> Line 36: Please remove "No increased staining occurred." The meaning of this sentence is redundant with the previous one.
<Response> I have deleted that text.
<Point 2> Line 69: Please emphasize the limitations of chemotherapy and use this to highlight the value of ICIs.
<Response> The content you pointed out has been added starting from line 69.
<Point 3> Introduction: You mentioned the studies by Grillo and Engel, which demonstrated a decline in the immunoreactivity of markers with prolonged storage time. Please emphasize the adverse consequences of this decline in immunoreactivity for these markers, such as how many patients might be misdiagnosed as a result.
<Response> Regarding the disadvantages of false negatives, additional text has been added to lines 106-109.
<Point 4> Line 119: Please add the number of cases.
<Response> We added the number of cases to line 126.
<Point 5> Line 202: It is quite rare to see a table cited in a title. Please correct this.
<Response> We have made the correction.
<Point 6> Please change the "p" used to indicate significance to italics.
<Response> We have made corrections throughout the entire text.
<Point 7> Please add spaces before and after the equals sign.
<Response> We have made corrections throughout the entire text.
<Point 8> Please check the formatting of the table text, such as capitalizing the first letter of "total."
<Response> We have made the correction.
<Point 9> Figure 1: The figure caption is not specific enough. Please provide a detailed description of what this survival curve represents.
<Response> We have changed Figure 1 to Figure 2 and revised the description accordingly.
<Point 10> Please change the font of the figure to Times New Roman.
<Response> We have changed Figure 1 to Figure 2 and made the correction.
<Point 11> Please review the entire manuscript, as I noticed some punctuation marks are not used correctly, particularly the frequent omission of periods at the end of paragraphs.
<Response> We have made the correction.
<Point 12> Lines 223-227: The meanings of these two sentences are similar. Please delete one
<Response> That section describes the efficacy of preoperative chemotherapy and survival outcomes for PD-L1-positive patients, respectively. Since we did not consider it to be the same content, we did not make any revisions.(lines 264-270)

Reviewer 3 Report
Comments and Suggestions for Authors
Beautiful work uprising an important question of molecular testing standartization and reliability associated to "ageing" paraffin blocks. The paper is written in a logical manner and structured well.
The abstract is coincise and perfectly explains the conducted study.
Introduction is clear and informative.
- In first paragraph (lines 48-62) I would mention more references between the sentences, as important statistics is being discussed, as well as the statements in lines 66-84 should be supported by the references.
Materials and methods well described in a reproducible manner, IHC protocol is very well described.
The Results are repsented in a coherent and structured format, Table 1 looks great and enhances the readability, but maybe could have an expanded tilte.
In paragraph 3.3. Change in PD-L1 Status After Storage it would be interesting to add a comment on % of decreased staining (>/< 10%) thus rasing a question (and maybe continued in Discussion) if older samples could potentially deprive patients from another therapy option, when newer samples not available.
Overall for me it has been an enjoyable, well-structured, readable and informative paper.
Thank You.
Author Response
To Reviewer 3
Thank you very much for your valuable comments. We have revised the manuscript in accordance with your suggestions.
<Point 1> - In first paragraph (lines 48-62) I would mention more references between the sentences, as important statistics is being discussed, as well as the statements in lines 66-84 should be supported by the references.
<Response> As you pointed out, I have revised it by adding the references.
<Point 2> The Results are repsented in a coherent and structured format, Table 1 looks great and enhances the readability, but maybe could have an expanded title.
<Response> As you pointed out, I have made the correction.
<Point 3> In paragraph 3.3. Change in PD-L1 Status After Storage it would be interesting to add a comment on % of decreased staining (>/< 10%) thus rasing a question (and maybe continued in Discussion) if older samples could potentially deprive patients from another therapy option, when newer samples not available.
<Response> As shown in Figure 1, we have added specific photographs illustrating changes over time. Additionally, we have noted in the conclusion and introduction sections of the Discussion that false negatives may deprive patients of treatment options.

Round 2
Reviewer 2 Report
Comments and Suggestions for Authors
Approved